# A Qualitative Exploration of UK Leisure Horse Owners’ Perceptions of Equine Wellbeing

**DOI:** 10.3390/ani12212937

**Published:** 2022-10-26

**Authors:** Rebecca Smith, Tamzin Furtado, Charlotte Brigden, Gina Pinchbeck, Elizabeth Perkins

**Affiliations:** 1Institute of Infection, Veterinary and Ecological Sciences, University of Liverpool, Leahurst Campus, Chester High Road, Neston, Cheshire CH64 7TE, UK; 2Equine Department, University Centre Myerscough, St Michael’s Road, Bilsborrow, Preston PR3 0RY, UK; 3Institute of Population Health, University of Liverpool, Waterhouse Building, Block H, Pembroke Place, Liverpool L69 3GF, UK

**Keywords:** equine, quality of life, welfare, wellbeing, horse-human relationship

## Abstract

**Simple Summary:**

Optimizing equine wellbeing is of concern to horse owners and key stakeholders in the equine sector. However, the language horse owners use to discuss wellbeing is not well understood and terms such as quality of life and welfare are often used interchangeably. Little is known about how those providing day-to-day care of horses use terminology, or what this means for wellbeing assessment. This study collected qualitative data from focus group discussions with UK leisure horse owners. Analysis identified that horse owners did not clearly delineate between different terms. Individually constructed equine wellbeing assessments were ongoing, and shaped by factors such as the horse’s purpose and owner’s ideas about good horse care. Strategies to support owners and improve communication must pay attention to the dynamic and contextualised nature of horse owners’ experiences.

**Abstract:**

Human assessment of equine wellbeing is fundamental to ensuring the optimal care of domestic horses. However, terminology associated with wellbeing is still not fully defined and there are currently no validated quality of life (QoL) assessment tools. Furthermore, little is known about what equine wellbeing or QoL means to horse owners, or how their beliefs impact on the management decisions they make for their horse. This study sought to establish how UK leisure horse owners use wellbeing-related terminology by exploring their accounts within a focus group setting. Four online focus group discussions (FGD) were held and qualitative data were collected. FGDs involved a semi-structured discussion, followed by a group activity to compare seven equine wellbeing-related terms of interest introduced by the facilitator. The collected data were analysed using a constructivist grounded theory approach, and also by content analysis, to examine the frequency and subjective meaning of the terms of interest. The results showed that horse owners did not clearly delineate between different terms, rather, they used the terms in the context of their own assessments of their horse. The meanings assigned to what owners experienced with their horse were individual and subjective, shaped by past experiences, relationships with their animal, and peers or social groups. This individualised construction of equine wellbeing impacted on the meaning conveyed when using wellbeing-related terminology. In this study, we extend the literature on equine wellbeing terminology usage, and highlight differences between the academic literature and the real-world experiences of horse owners.

## 1. Introduction

In the field of equine welfare science, strategies to improve equine lives have focused on enhancing human assessment of the horse, on the premise that better recognition of poor wellbeing could enable horse keepers to improve welfare outcomes [1]. Scientific understandings of equine health and welfare have undoubtedly increased [2]. There is now increasing focus on more holistic approaches to animal wellbeing assessment which value and attempt to measure positive experiences which are relevant to that individual [3]. Contemporary understandings of welfare focus on individually meaningful, species-appropriate experience in relation to nutrition, physical environment, health, and behavioural interactions, which culminate in an animal’s mental state [4]. With this approach to animal welfare, the concepts and associated language around equine wellbeing are changing. In the UK, research around human perception of equine wellbeing has focused on equestrian professionals, including various “stakeholders” [5] and “professional riders” [6]—despite the majority of horses in the UK being cared for by leisure horse owners. Little is known about the way in which those caring for horses on a daily basis conceptualise equine health and wellbeing. Therefore, this study sought to explore the experiences and language used by leisure horse owners (hereafter referred to as ‘horse owners’), in order to understand how the concept of equine wellbeing fits within their daily lives.

Research has found that horse owners and veterinary surgeons (veterinarians) use terminology such as quality of life, wellbeing and welfare when discussing topics including equine obesity, ageing and end-of-life care [7,8,9]. At times when views of a horse’s welfare or QoL were divergent, communication between veterinarians and horse owners/carers was reportedly challenging [10]. A recent study of horse owners [11] found that people held beliefs about what these concepts meant to them in theory, however, in relation to their own horses, they reported that these principles were not always achievable. It is unclear how the terminology around equine wellbeing is used, what it means to those responsible for everyday horse care and how people communicate these beliefs to others.

While the meaning of the word “welfare” has received extensive attention in scientific writing [3,12,13], other wellness-related terminology are still not fully understood, and the crossover between terms is apparent [14]. Indeed, in writing this paper, it is difficult to discuss such concepts related to equine wellbeing without using one term to describe another. The use of such language, and the meaning it conveys, has been highlighted as important in the field of equitation science [15], yet this has not been well explored in terms of everyday usage.

To date, there are no validated Quality of Life (QoL) assessment scales for horses, and so assessment is encouraged through the use or adjustment of existing welfare assessment tools and pain scales [5,16,17]. Recent research [6] has reported that formal QoL frameworks were not used by competition horse owners who opted instead for more informal ways of understanding their horse. Health-related and holistic QoL assessment tools are available for small companion animals, yet despite their availability, uptake is poor, and challenges around how, by whom and in what way QoL is best assessed remain [18].

Despite the attention and care devoted to leisure horses, health and wellbeing issues are widespread: these include unrecognised pain, obesity and delayed euthanasia [19,20]. Several studies have explored how horse owners conceptualise, recognise, and make decisions around issues which can negatively impact horse wellbeing. These include decision-making around health concerns such as colic and obesity, as well as approaches to older horse management and health care [9,11,21]. Early detection of health problems may prevent more serious issues from developing, but in order to detect these problems, it is important that horse owners have an understanding of their horse’s baseline wellbeing. As scientific research has focussed on specific health concerns there has been a paucity of research around how horse owners conceptualise, monitor, and manage good health and wellbeing in their horses—or the language they use to describe this.

The aim of this study was to establish how leisure horse owners conceptualise horse wellbeing and use wellbeing-related terminology. Using a qualitative approach, this study sought to explore horse owners’ use of equine wellbeing terminology, to understand the relationships between the terms of interest and to observe the process of collective understanding in a group context [22]. 

## 2. Materials and Methods

This paper reports on qualitative data collected from four focus group discussions (FGD) held from February to April 2021. Participants who identified themselves as leisure horse owners were recruited using targeted social media advertisement and gatekeepers known to the research team. Initially, all participants who applied to take part in the study were included; however, no male horse owners had replied, and hence purposive sampling with an advert specifically seeking male participants was employed. Participants had varying degrees of equine experience, kept horses for different leisure roles (ranging from amateur competition through to retirement), and were from across the UK and a range of ages. Brief participant information is given in Table 1 below, though locations and ages are not provided so as to maintain anonymity. Each FGD consisted of 3–7 participants with 21 participants (14 female, 7 male) in total. All participants gave their informed consent prior to the focus group and participants were able to cease participation if requested (none requested this).

FGD were held online using Zoom and were approximately of 90 min duration. RS and TF facilitated the semi-structured FGD. Following an explanation of the study’s purpose and planned methods, participants introduced themselves to one another. The sessions then consisted of two parts which aimed to first elicit participant’s own experiences of thinking about wellbeing and explore the language used in their spontaneous speech, then to explore the terms of interest comparatively. To achieve this, an initial discussion using a topic guide (e.g., “could you tell us about a time when you have thought about a/your horse’s wellbeing?”; see Appendix A), was followed by a group activity involving the comparison of pre-selected terms of interest displayed on a virtual whiteboard. Terms relevant to equine wellbeing and welfare were selected based on; their use by horse owners in previous equine studies [9,23,24], equine media, welfare science [3,25], and personal experience of the research team members. Terms comprised; quality of life, welfare, life worth living, wellbeing, happiness, good life and best life. Participants were asked to describe and discuss what the terms meant to them, and to identify any relationships between the terms.

Data were audio-recorded, transcribed verbatim, and anonymised by altering any identifiable data such as horse names, locations, or veterinary practices. Transcript accuracy was checked whilst listening to the audio-recordings and this aided familiarity. Data were analysed by three researchers (RS, TF, CB) using a constructivist Grounded Theory methodology [26]. This approach adopted a social constructionist epistemology and acknowledged the researchers’ role in the construction and interpretation of data [26,27].

One FGD transcript was selected for open coding by RS, TF and CB, who initially coded the transcript individually by assigning iteratively developed codes or “labels” to units of speech. Initial codings were jointly reviewed and codes amended to ensure consistency. Further analysis involved constant comparison, through which data were compared and codes were reviewed and refined. This involved examining language and the context of its use, as well as how participants interacted during discussions. During this process, codes were grouped into larger categories and conceptual themes were developed. Conceptual models were developed as a way of analysing relationships between data. Using theoretical sampling, we recruited participants for a male-only fourth FGD in order to explore whether wellbeing assessment was different in the context of these horse-human relationships.

In addition, a content analysis of the data was undertaken to enable direct comparison of the frequency and use of terms. Results are discussed in Section 3.2

Whilst reflecting on our own roles as researchers we felt it was important to consider our experience of owning or caring for leisure horses and position as equine welfare researchers. We met regularly to discuss our own interpretations of meaning in the data and reviewed ongoing analysis. Upon collection of data from the four FGD we considered our analysis to be conceptually rich and we describe this as reaching theoretical saturation [28]. All quotes included in this paper have been anonymised for presentation. 

## 3. Results

This study identified themes which were developed into a conceptual model to illustrate how individuals viewed a horse’s wellbeing: demonstrating the ways in which this was shaped by their past experiences, social influences and their relationship with their horse. These individualised constructions of wellbeing reflected differences in how participants constructed the relationship between health and wellbeing.

There was diverse use of terms such as wellbeing, welfare and happiness across the groups, but despite this, there were no examples of confusion around item meaning between participants. Participants demonstrated a tacit understanding of the concepts being used even though the same words were used in different contexts. The terms most often used in spontaneous speech by participants to describe a horse’s state were welfare, and happiness. Each of the terms of interest were used with a variety of meanings, incorporating a great deal of crossover. This highlights the semantic fluidity of the language used to describe animal wellness, with meaning being constructed through the context in which the language is used.

The results section will focus first on how participants’ understandings were generated, before focussing on the linguistic constructs explored in this study. 

### 3.1. Contextualised Wellbeing Assessments

Participants’ understanding of horse wellbeing were generated through ongoing informal assessments as part of everyday horse care. When asked to tell the group about a specific time when they had thought about one of their horse’s or any horse’s wellbeing, participants said “*everyday*” or “*all the time*”.

Four major themes were identified as shaping participant’s individualised understanding of wellbeing, which in-turn shaped how they made more formalised judgements. These were ‘principles of “good” horse care’, ‘modifying outside influences’, ‘ways of knowing’ and ‘assessing the individual horse’ (Figure 1).

At the broadest level, ‘principles of “good” horse care’ were particular ways of managing a horse that participants believed were most aligned with achieving optimal wellbeing. ‘Modifying outside influences’ were factors which informed these principles. These factors generated understanding around what was necessary to provide for a horse, without needing to be situation-specific.

As owners considered more specific, situational wellbeing assessments, ‘ways of knowing’ provided a means for owners to make decisions about what was “right” or “wrong” for an individual horse. These ways of knowing could shape the subsequent perception of a horse’s wellbeing and were closely related to an owner’s relationship with a horse. Through a person’s interactions with their horse, their principles were constantly adapted and updated in line with their changing knowledge about, and relationship with, the horse. Therefore, the process of caring for a horse, as well as wider societal change, shaped the lens through which assessments were made. Consequently these horse owners might have an entirely different perspective on the wellbeing of their individual horse, based on individual experiences and ideologies, compared to the assessment performed by another person, such as a veterinarian or another horse owner.

As these ongoing wellbeing assessments were part of owners daily interactions with their horse, they formed a basis for making decisions about care. In this way, care practices comprised an ongoing, iterative cycle of observation and management adaptation.

Each of the four themes will be discussed in turn, beginning with the broadest level (principles of “good” horse care) and moving to the most specific (assessment of the horse as an individual).

#### 3.1.1. Principles of “Good” Horse Care

Participants’ understanding of care practices that aligned with ‘responsible’ horse ownership translated into judgements about what was “right” or “best” in terms of horse care, e.g., not riding a horse before physical maturity, keeping horses barefoot (without shoes), or not selling horses. Principles were constructed, such as discourse advocating keeping a horse “naturally”. Where participants were aligned with these principles, this became part of their horse-keeping identity:


*“I like them to have as natural a lifestyle as possible… Everyone is out 24/7. There’s no rugs, there’s no clipping. The ones that are ridden are bitless. There’s no shoes. Natural is just my thing.”*
(focus group 4)

As participants communicated with one another regarding these ideologies it was evident they were used as a way of expressing their own sense of self. For example, keeping horses barefoot was constructed as a major part of ones’ identity as a horse owner, and a shorthand for an ideology around “naturalness”; a discussion which formed a core part of focus group three.

Individuals understood that a horse had certain needs, both in terms of provisions and activities involving the horse. However, the practicalities of this were closely linked to a horse’s given purpose, and this impacted on beliefs around the provision of turnout, companionship and ridden exercise:


*“He was a show horse. So they would want to keep everybody separate”*
(focus group 3)


*“I had a horse who would have been quite happy as a pasture ornament. But for him to be a respectable member of society he wasn’t allowed to do that, he needed a job”*
(focus group 2)

Beliefs around how a horse should ideally live, and what compromises were acceptable, were understood to differ in line with discipline norms. Therefore, what one group may consider to be a concern, another may not:


*“I’ve known people tell me about a quarter of a million pound Olympic dressage horses, which are psychotic, because they can only be ridden by one person, they can only be ridden in a strict routine, from the stable to the arena. Never ridden outside, never goes into a paddock. So you’ve got these sort of quality of life.”*
(focus group 4)

These constructs were also expected to change in line with increasing age, requiring an individual approach to wellbeing assessment:


*“those words, good life, best life or life worth living, are things that we impose on our horses or animals, from a perspective, even our relatives. That’s where our own sort of personal views and values come in, to make that judgement on what we feel a horse’s quality of life is going to be like. That quality of life is going to change at the various stages of their life and how old they are.”*
(focus group 4)

Social norms linked to specific disciplines or horse use shaped individual owners’ beliefs and values relating to what equine wellbeing meant, and what was necessary to ensure it. However, there was acknowledgement by some participants that views on this varied:


*“if you had a young horse that then sustained an injury that meant that it wasn’t going to be a racehorse, like it was expected to be. But it’s still going to be able to be a good companion horse, someone might feel that actually that’s not a good quality of life for that horse that had the potential of being something amazing and might take a different view on its wellbeing”*
(focus group 4)

When conversations were theoretical (such as when discussing someone else’s horse, or a hypothetical situation), these principles were strongly relied upon. For example, when applied to weighing up equine QoL or potential euthanasia decisions, participants also made judgements about horse-keeping practices which did not align with their own principles. One participant described drawing on these principles when considering situations such as the need to box rest a horse for a lengthy period of time:


*“if it had been my pony I think I would have had her put to sleep, because it just felt cruel that this pony was literally living in a box. And I don’t think that is right, personally.”*
(focus group 3)

As well as questioning others’ practices, one participant also spoke about their own experiences of being reported to an animal welfare charity, despite believing their horses were being well-managed. As each individual horse owner constructed their own unique ‘principles of “good” horse care’, and assessed a particular horse in the context of it, in reality, perceptions of equine wellbeing differed between participants.

Similar inter-participant differences have been revealed about the moment at which euthanasia was deemed appropriate. Despite end-of-life decision-making being constructed by participants across the focus groups as a part of responsible ownership, the point at which wellbeing was considered compromised enough to warrant euthanasia, varied from person to person, and could be reconstructed at later time points.

#### 3.1.2. Modifying Outside Influences

In this study participant’s horse-keeping environments included livery yards, retirement livery, private or home premises. Many participants were conscious of the influence of peers within their horse-keeping environment, particularly when individual beliefs on appropriate care conflicted with peers, leading to differences in perception of an individual horse’s wellbeing:


*“Some people on the yard think we’re ridiculous, keeping her going, given that we don’t use her, other than walking her out every day and so on”*
(focus group 2)

These environments created the context in which relationships with, and care of, the horse took place. Decisions about where a horse was kept were multifactorial, and owners often had to make compromises regarding their horse’s lifestyle when making yard choices. On livery yards the individual horse’s care was situated alongside the needs of all the other horses on the yard, meaning that the owner’s preferred options (for example, full-time turnout) might not have been possible. As choices made by owners were based on their principles of ideal horse care—for example, the desire to offer daily turnout as a perceived core component of wellbeing—this sometimes led participants to choose yards with turnout which were impractical in other ways. Some owners had moved premises, some preferring to move from livery yards to private/individual premises in order to achieve more control over their horse’s daily care and offer a lifestyle which contributed to a perceived higher level of wellbeing:


*“having to compromise when farmers or livery owners, yard owners don’t want to mess with their fields, well, do you know what, I don’t care about messing up a field. The horses’ wellbeing comes first for me, and that’s why I ended up just renting a field so that I got that choice, and I’ll never go back.”*
(focus group 1)


*“I have moved yards before because they didn’t have enough turnout, for instance, and my one needs to be out or she is not healthy or happy, and she gets those stereotype behaviours, and it is just not great”*
(focus group 3)

When considering their horse’s wellbeing, owners’ views were sometimes informed by independent research as well as professional advice. For some participants, the need for social support which aligned with their own views meant advice was sought online, where groups of horse owners could come together around a locality (e.g., local horse owners) or principle of horse care (e.g., keeping horses barefoot). In these groups, principles of care were socially reinforced, leading to owners forming communities of practice around perceived methods of improving equine wellbeing. For example, the following participant had started a new online group for owners of horses who had 24/7 turnout:


*“We’ve got over 8000 members now. And so many people come on there and go, “I am so glad to have found this group because on my yard everybody’s saying ‘Why isn’t your horse wearing a rug, why isn’t your horse clipped, why doesn’t your horse come in at night, why are you leaving him out in all the rain?’” You know, and they’re so pleased to find a group that agrees with everything they say because most of the yards are still treating horses like they were dogs or humans.”*
(focus group 1)

These groups contributed to and supported owners’ perceptions of what constituted equine wellbeing by providing social validation, often in situations (as in the example) where others had contested those views.

Veterinary advice was infrequently discussed in this study. Advice was reportedly sought when deemed necessary or relevant to an issue of concern. For some participants, veterinarians could provide specific additional knowledge:


*“I don’t have my vet’s skill, tools or ability to assess their physiology or to see what’s going on.”*
(focus group 4)

However, there was some scepticism over seeking veterinary advice. For example, perceptions of a limited knowledge of an individual horse, unsuitable husbandry advice, or inappropriate guidance around diagnostics or treatments, impacted on how veterinary knowledge was perceived:


*“And even my vet, with the last laminitic episode that my quarter horse had, said you need to only feed her so much. She’s not overweight. It wasn’t that, it was the type of forage, so he was completely off the ball and he’s lost a bit of credibility with me but that’s another issue.”*
(focus group 1)

#### 3.1.3. Ways of Knowing

Owners employed three strategies in order to make judgements about a horse’s wellbeing. These included ongoing learning about the horse, monitoring a situation over time, and using heuristics. Owners combined these strategies to make decisions, though in some situations one strategy could be more strongly relied upon than others. Although these strategies or ‘ways of knowing’ could be applied to any horse or situation, relationships with individual horses were needed to make the most appropriate judgements.

Ways of Knowing: Ongoing, Experiential learning

The ability to understand horse wellbeing and horse care was constructed as an ongoing journey. Participants suggested that they were continually updating their knowledge and practices. Ongoing learning was a result of increasing their knowledge (for example, social learning from yard peers or social media), and experience. Consequently, owners often viewed their past actions as examples of poor horsemanship which resulted in compromised wellbeing in their own horses:


*“You learn something that works and then you are like ashamed of what you used to do.”*
(focus group 3)


*“As I’ve tried to find out more and more, I’m still aiming for my horse to have its best life for each one of them. I’m probably not perfect, well, there’s no probably, I’m not perfect, I’m still not there but I know that my horses have a much better welfare standard than what they would have been 15 years ago with what I’m doing and how I’m keeping horses. They’ve got a much better quality of life for sure”*
(focus group 1)

Participants described scenarios in which, on reflection, their equine-related knowledge had increased over time. For example, one participant described his increased confidence in monitoring his partner’s horse’s welfare, as a result of his increased time spent with, and therefore better knowledge of their horse:


*“What I’m getting from a lot of the gents here is that experience over many years has led them to be able to recognise that. At first, maybe in the first few months of me looking and caring after the horses at the time, I didn’t recognise those things because I wasn’t sure what I was looking for. But now, three years down the line, I start to feel like I could pick up things like lameness or just a general, being subdued, whilst in the stable etc”*
(focus group 4)

Along with scenarios in which experiential knowledge was applied to the judgement of equine wellbeing, some participants shared how end-of-life decisions would likely be altered by knowledge from previous experiences. Past experience could form a frame of reference within which judgements about current situations could be made:


*“If she ever had that again, I don’t think we’d put her through it. So, welfare and wellbeing, it obviously worked out she had another three years fine, but if it ever happened again, I don’t think we’d do that again, would we?”*
(focus group 2)

Ways of knowing: Monitoring over Time

Monitoring changes over time allowed horse keepers to determine whether the horse’s wellbeing was subject to change; change was considered indicative of a potential problem which could warrant attention:


*“I think if anything changes from the normal. That’s what I would indicate as an issue, if their behaviour changes.”*
(focus group 2)

Participants described that all horse interactions involved some level of (often unconscious) monitoring of the horse’s wellbeing over time:

“Facilitator: *Would you be able to tell us about a specific time when you’ve thought about your horse’s wellbeing?*

Participant: *About every day. (Laughter) …… you’re constantly thinking about the welfare and wellbeing of your horses. It is a constant”*(focus group 1)

This strategy could be used for any horse seen over a period of time; some participants spoke about monitoring and assessing the wellbeing of horses owned by others but which they were in contact with regularly. One participant was an instructor to other horse owners and was able to monitor their horses during her teaching sessions, while others spoke about observing the wellbeing of horses on their yards, or nearby fields over time. Contrastingly, several owners described the ongoing monitoring of their horse’s wellbeing by livery yard employees, particularly where a horse was on full livery.

Interestingly, monitoring for changes in all circumstances was informal, with no instances of participants describing keeping registers or diaries of behaviour or physical health (although they were not specifically asked about this).

Ways of knowing: Heuristics

Heuristics are a “decisional short cut” [29] in which decision-making happens as a result of known rules of thumb. In this instance, horse owners used heuristics as a strategy to assess what was right or wrong about a situation, through their combined principles of horse care and experience. Together, these factors enabled participants to make judgements about practices which were perceived as acceptable or unacceptable in relation to equine wellbeing:

[about a horse on long term box rest] *“I felt that wasn’t a quality of life. I thought it was a horrendous life for that horse. It existed.”*(focus group 3)


*“in my head, I’m very clear that, if I can’t get him right, he is going to the big paddock in the sky. Because I’m relatively young and relatively fit and if I won’t sit on it, nobody else is going to.”*
(focus group 2)

In both instances, and many others in the data, participants described principles they used in order to make wellbeing associated judgements (e.g., “*box rest is not good quality of life*” and “*I would not sell a difficult horse*”). 

Some participants also described heuristics around age-appropriate activities, which led them to consider that certain lifestyles might be more or less suitable at different ages:


*“Yes, because they’re the same as humans, they learn to understand more about how life is. You wouldn’t expect a yearling to spend its days working, even if it was work that it was physically capable of, whereas you get a horse 10, 12 years old, they’re in the habit. They can spend more time in the stable, they can spend more time being ridden without being stressed by it.”*
(focus group 1)

Interestingly, there were many instances throughout the data in which people felt it necessary to diverge from the heuristics that they relied upon in other situations. For example, participants who preferred not to stable a horse also described times when the situation necessitated stabling, or participants who preferred horses to be kept in company described keeping a horse in isolation:


*“I have got a little field near where I live, so I have kept her in there for quite a few years on her own. I don’t think it is ideal having them by themselves but needs must. But she does get out a lot. She goes out riding a lot with other horses. So I don’t feel that she was particularly unhappy in her little paddock by herself.”*
(focus group 3)

Participants’ heuristics were therefore broadly applied but could be flexible when an individual situation demanded a compromise.

#### 3.1.4. Assessing the Individual Horse

Participants were clear about the importance of knowing and assessing their horse as an individual in order to make more accurate distinctions about his or her wellbeing. Unanimously, discussions between participants were based on an agreed underlying assumption that all horses have individual personalities, likes and dislikes, and that owners learnt about the individual horse through their ongoing relationship, and subsequently use this knowledge to assess his or her wellbeing and inform care choices.

Knowing the Horse as an Individual

Knowledge of the horse’s individual personality and physical wellbeing was generated over time through the horse-human relationship. This included an interpretation of the horse’s needs and known likes and dislikes. For example, one participant described that her mare “craves” human interaction, despite not wanting interaction with other horses:


*“I think human interaction. I know that [Pearl] craves it, doesn’t she? She doesn’t really like other horses. She’s a mare and a chestnut mare, typical, she’s a right bully. And I’ve seen her—I’ve had her 21 years and I’ve seen her groom another horse twice. But she will crave human interaction.”*
(focus group 2)

This knowledge, reinforced over the 21 years of ownership, translated into an adapted model of care for Pearl, who was kept alone but with plenty of human interaction. 

Similarly, other owners interpreted their horse’s needs and preferences, such as turnout and exercise requirements:


*“he likes a job. If he’s in the field 24/7, if there aren’t enough people to cause trouble with, then he starts causing trouble in other ways”*
(focus group 2)

Such knowledge was built through the horse-human relationship over time, leading to a deep understanding of that individual which was part of ownership. As one participant commented “*no-one knows them like you do*”.

Assessment Tools (e.g., Behaviour, Physical, Listening to the Horse)

Knowing the horse intimately meant that horse owners could use different components of their understanding of the individual animal in order to assess the horse’s current state of wellbeing, establish appropriate individualised management, and monitor changes over time. The knowledge of the horse therefore becomes a type of social capital, respected by other equestrians as unique knowledge built between that horse and owner as a result of their ongoing relationship. Participants described monitoring behaviour, demeanour, physical wellbeing, or “listening” to the horse as tools for evaluating and managing the individual horse.

Assessing the horse’s physical wellbeing was considered to be relatively straightforward and thus conducted on an ongoing, informal basis alongside the horse’s daily care, and led to individualised management decisions. For example, participants monitored their horse’s physical condition in order to decide how to feed, rug and exercise their horse to optimise health. One participant discussed how her assessment of individual physical needs led to her rugging her horses in different ways:


*“I’ve got two native ponies, they’re both trace clipped, they haven’t got a rug on. Right? I’ve got a Lusitano imported, if it rains heavily in the winter, she is shivering because she’s got no waterproofing in her coat. So she’s got a full coat and has to wear a rug and the other two are trace clipped and you never see them shiver. And it can depend on the individual horse as well, not just the breed. I know other people who’ve got Lusitanos who are like yaks in the winter.”*
(focus group 1)

Importantly, this participant described that her decision was based very much on the specific horse in her care; she did not use other heuristics, such as breed, as a proxy for individual knowledge.

Behaviour and demeanour were also used to assess individual wellbeing; for example, one participant described a friend whose horse’s demeanour changed as a result of a management intervention (moving to individual turnout):


*“she had a really arthritic mare, who just couldn’t get out of the way of other horses. And she said that she noticed as soon as she went into individual turnout she just seemed so much more relaxed, because she wasn’t constantly worrying about having to get out of the way of other horses.”*
(focus group 3)

The change in demeanour of this horse enabled the owner to re-evaluate the horse’s needs in relation to socialisation, despite the fact that all participants agreed that group turnout was generally the ideal in most instances.

By reading their horse’s behaviour, participants could also “listen” to the horse, which sometimes extended to allowing the horse to make choices about how their own care was navigated. A common example of this was in relation to the extent to which horses “wanted” to be turned out compared to in stables, which was often viewed as dependent on the individual personality of the horse:


*“it is just again based on my horse’s personal preference. Like my red mare doesn’t want to be out 24/7. I tried that and she hated it. She likes to come in for a bit to get a break from the wind and the rain. The other one is kind of take it or leave it with turnout.”*
(focus group 3)

“Listening” to the horse sometimes transcended other types of knowledge; for example, one participant described that reading her horse’s behaviour led her to eschew her traditional knowledge and learning:


*“that point was the first time I listened to my horse and said, Okay, whatever I know and whatever I’m being told and whatever is traditional horse-husbandry isn’t actually working for you, so we’ll try a different path”*
(focus group 2)

Issues with Wellbeing Assessment Based on Individual Knowledge

While participants largely relied on the assumption that experiential individual knowledge was an essential component of wellbeing assessment, there were examples in the data of retrospective discussion about “knowing” the horse being inadequate for providing an accurate picture of wellbeing. For example, horses who had behaved in ways which were indicative of pain throughout the duration of the horse-human relationship, could have those pain-related behaviours overlooked, attributed to a part of that horse’s individual personality and shaped by the horse’s assigned purpose. For example, the following case study describes an owner who felt she knew her horse “inside out” “like an old pair of slippers”, overlooked behavioural indicators which were indicative of pain because she assumed they were simply a feature of being a “typical thoroughbred”:


*“my black horse, the one who started all this off, I always thought he was just a typical thoroughbred. He hated being groomed, he was really flighty, whatever, whatever. He wasn’t particularly affectionate, he’d come from a competition yard, he didn’t know what an apple or a carrot was. He had had a really bad start in life. So, I just thought that was him. And then as I was going down the whole rabbit hole of natural husbandry and barefoot care and stuff like that, I started reading about ulcers. Anyway… after 10 days on Omeprazole, it was like he’d had a character transplant……So, this horse that I had had for 10 years, that I thought I knew- Well, I did know inside out, that was like an old pair of slippers, suddenly turned into this affectionate donkey that couldn’t get enough of being groomed and used to follow you around the field…. suddenly I was like, “Shit, this horse I know really well is actually someone else entirely.”*
(focus group 2)

The transformation from a “flighty, typical thoroughbred” to “affectionate donkey” caused this participant to question and re-evaluate her knowledge of the horse as an individual, and reattribute aspects of his behaviour. Although fundamental to enacting care, the meaning of this owner’s knowledge was reconstructed in a new context. By incorporating new knowledge and different management approaches, here the horse-human relationship has been transformed.

### 3.2. Linguistic Representations of Wellness 

Descriptions of a horse’s state included reflection on a horse’s physical and mental wellness; this was expressed variously through the terms; “welfare”, “wellbeing”, “happy” and sometimes “quality of life”. Results of a content analysis of the terms of interest are presented below in Table 2. Terms could be used almost synonymously, and were sometimes employed hand-in-hand by juxtaposing two terms such as “welfare and quality of life”:


*“you’re constantly thinking about the welfare and wellbeing of your horses. It is a constant and it probably changes with the weather”*
(focus group 1)


*“I think that is a nice quality of life. She seems to be happy. So I am just going to keep doing it until she can do no more.”*
(focus group 3)

Although the non-specific usage of these terms (as above) was common, on other occasions participants used the terms comparatively to express and differentiate their understanding of equine wellness states. The following section will focus on those differentiated meanings. The diverse usages of each term are displayed in Table 2, Results of a content analysis of the terms of interest.

#### 3.2.1. Welfare and Wellbeing

The term “welfare” had the most diverse usage, being variously described to mean general equine physical and mental wellness, a general sense of provision (for example “*welfare is something that we give to the horses*”), a minimum acceptable environment (for example, provision of food, water and bedding), minimum acceptable equine experience (for example, the extent to which a horse can live with pain), neglect, or positive equine experience.

Participants noted that this term could be “loaded” due to its usage in relation to legal cases, and thus had varied meanings:


*“you can have welfare cases where horses are neglected and abused and emaciated. Well then you’ve got welfare, whether you are riding it and it’s got brushing boots to cover up sores or saddle galls or something like that. Welfare, in what context is the question? It can go in many, many directions. It could be rather sublime, or it could be extreme.”*
(focus group 4)

Participants used the term in conversation with one another with reference to this broad range of meanings; for example, one participant with a legal background familiar with the legal sense of “welfare”, nevertheless described their own horse’s “*welfare issue*” in relation to having a hoof abscess when her shoes were removed—a relatively minor, short term and common problem. Other participants also referred to “welfare” in relation to relatively minor issues which were being constantly monitored:


*“we’ve been monitoring our guys welfare. My wife’s horse is coming back from a tendon injury that he did last summer, so we’ve been doing some rehab with him.”*
(focus group 4)

As identified above, welfare could denote the delivery of minimal standards. For example, one participant described ponies kept in a barren environment:


*“I see people there every day, so they get water, but they don’t look well cared for. But I know those people are doing the best they can, and I know that they’re there every day, they have fresh haylage, fresh water. But for me, I don’t think it’s a good life for the horses. But I do think that their welfare has been met. But is it the best life? No.”*
(focus group 2)

In this instance, “welfare” reflects the basic and suboptimal provision for the ponies. On other occasions, participants’ linguistic usage reflected different meanings, with “welfare” denoting a minimum standard of care, and “wellbeing” associated with affective state:


*“for me, every day with a horse is a question, “How are you? How are you feeling? What can I do for you today?” And some of that is wellbeing. But the stuff like, should horses be in a paddock on their own, should they be confined for 23 h a day? All of that stuff, that’s welfare. It’s not whether you destroy them or not because they’re limping down the road, it’s every decision you make for your horse when you’re keeping them should be a welfare issue.”*
(focus group 2)

#### 3.2.2. Quality of Life (QoL)

“Quality of life” was most commonly used in relation to considerations over a minimum acceptable life experience for a horse, and hence its use was often described around euthanasia decisions:


*“It didn’t want to do anything. It came out of the field. It went in the stable. It went back. To me, I felt that wasn’t a quality of life. I thought it was a horrendous life for that horse. It existed. And it didn’t look like a happy horse... I just felt it was cruel, again, keeping it going”*
(focus group 3)

In this extract, QoL is considered in relation to the horse’s lack of “happiness”. QoL was also used to relate to positive wellbeing more frequently than most other terms, again highlighting the diverse usage of each item of interest.

#### 3.2.3. A Life Worth Living, Good Life, and Best Life

The following terms were not generally volunteered by participants; they were infrequently used in the first activity whereby participants described their own horses, but were readily understood and discussed during the second task which involved comparing the terms. When prompted, participants used terms such as “a good life”, “a life worth living” and “best life” to refer to the horse-related outcomes of their care provision:


*“Racehorses, they don’t have a life worth living. They really don’t. They don’t have anything a horse needs, the top show jumpers and especially the top dressage horses, they don’t have the life worth living at all.”*
(focus group 1)


*“a good life and best life obviously aren’t the same thing, because a horse can have a good life but a horse’s best life, it certainly wouldn’t ever been ridden or has to do anything for humans”*
(focus group 1)

During this comparative exercise, a “life worth living” was considered a minimum acceptable standard of living, compared with a “good life” usually referring to an acceptable life, and “best life” to an optimum level of wellbeing.

“A life worth living” and to a lesser extent “best life” and “good life” generated additional discussion about the relevance of human judgement on equine wellbeing. For example:


*“I think the life worth living—who decides? It’s the same thing. I’m a cancer surgeon as well—who decides what life is worth living. I guess some of the decisions we make for our older horses, that’s what we’re doing every day, isn’t it?”*
(focus group 2)


*“Life worth living, best life, those are judgements made by owners, they’re not something that a horse is ever going to consider”*
(focus group 4)

#### 3.2.4. Happiness

Participants most commonly referred to happiness through its absence: they used the terms “unhappy” or “not happy” throughout their narratives to describe a general sense of equine malcontent, whether physical or psychological:


*“If the end goal is worth it, yes, you may inconvenience your horse by putting in a stable for box rest. It might not be particularly happy about that but if you know that, in the end, it will come good, then that’s worth it. Maybe it’s a bit unhappy for a few weeks, you try and do the best you can, entertain it.”*
(focus group 2)

The term “happy” was used in relation to positive equine affect only six times; instead participants predominantly used it in the general sense of wellbeing described above, or to denote a state of “contentment”, which was considered to better express equine experience compared to “happiness”:


*“I think if you want to know if your horse is happy, look for contentment. You know, are they chilled out when they’re just roaming around having a munch? And you can see horses that aren’t happy that are weaving, because they’ll weave in a field, they’ll chew the wood, they’ll chew the fence, they’ll not stand still”*
(focus group 1)

The above participant conflates a lack of happiness with behaviours which may be indicative of more severe or prolonged distress.

## 4. Discussion

This study represents the first attempt, to our knowledge, to identify and explore leisure horse owners’ conceptualisation of wellbeing, and use of wellbeing terminology, through qualitative methods. In doing so, we have highlighted the complexity of this topic: wellbeing-related terms are used to represent a variety of meanings depending on context. Importantly, it was not the words themselves that represented different levels or aspects of wellbeing to horse keepers, but the relationship with the individual horse in the context of that person’s experience and value judgements.

Despite participants using these terms to represent a variety of meanings, groups were able to freely converse around issues of horse wellbeing. In the study of human language (linguistics), the ability of people who are using different languages to understand one another is described as mutual intelligibility [30]. Our findings suggest that wellbeing language is similar enough to enable horse owners to comprehend one another even though they may attribute slightly different meanings to the words. Further research is required to explore this in the context of different groups conversing (e.g., horse owners and professionals) to examine whether this is also the case, or if the use of these terms contributes to communication difficulties.

Horse owners incorporated both physical and mental health when assessing equine wellbeing, and considered these as independent but intertwined; for example, they spontaneously discussed examples of horses who had good mental wellbeing despite physical health issues, and vice versa, yet considered that each impacted the other. Owners felt responsible to maintain an equilibrium between both equine mental and physical health, based around their knowledge of the individual animal (represented as capital) and their specific personality and care needs. In racehorse welfare, individualised care decisions were considered to be necessary for racehorses to live their “best life” [25] as compared with a “one size fits all” approach which provided minimum acceptable standards. In our study with leisure horse owners, it was implicitly assumed that horse owners would make decisions based on their knowledge of the specific owned horse, and adapt care accordingly over time. To provide a “best life”, owners felt they had to move beyond individualised care provision to provide optimised experience, though they felt this was sometimes unattainable because of the realities and compromises necessary in horse-keeping.

While owners considered mental health in their understanding of horse wellbeing, it was clear through the analysis that perceptions of wellbeing were primarily focussed on avoiding negative experience, rather than the promotion of positive experience. For example, the most commonly used expression of welfare (regardless of word choice) was a general absence of nothing being overtly wrong, often represented through the horse not showing signs of distress such as stereotypical behaviours or physical health issues. There was also an assumption that wellbeing could be a result of the provision of horses’ basic needs (bedding, socialisation, feed, and water for example). Combined, these results suggest that the levels of wellbeing horse owners consider acceptable may be more focussed on the avoidance of negative welfare indicators than the expression of positive ones. This has also been found in a previous study of stakeholders in equestrian sports [6], where there was a mismatch between the sporting bodies’ promotion of the idea of the “happy equine athlete” and participants’ expectations of horse wellbeing. Research has also found that horse owners are not always adept at identifying equine affective states [31], and our results clarify the need to take into account the spectrum of perspectives of wellbeing when creating resources aimed at broadening owners’ knowledge. Given that the move in broader animal welfare science towards positive welfare (rather than an avoidance of negative) is relatively recent [3], this suggests that horse keepers could require additional support with identifying positive emotion in horses, and considering equine lifestyles which will promote positive experiences.

None of the participants in this study discussed having used welfare assessment tools or checklists, and our interpretation of their descriptions of their thought processes suggested that their assessments were far more complex, nuanced and dynamic than wellbeing checklists provide. For example, although the participants discussed measurable concepts such as lameness, appetite, and interactivity which might be expected on a welfare checklist, they placed each of these items in the context of their ongoing daily assessments, heuristics, and knowledge of that individual animal. Similar findings were reported by farmers interviewed about their approach to treating lameness in dairy cattle, where participants relied upon being able to detect problems through daily routines and believed that objective tools such as mobility scoring would add nothing more to the detection process [32].

Despite the range of examples of welfare described across the focus groups, participants did not at any point describe wanting assistance in monitoring or assessing wellbeing. Reimagined welfare assessment support as a result of this study could incorporate an understanding of the dynamic, daily nature of the equine monitoring which horse owners undertake as part of their ownership responsibilities, as well as assisting in bringing to the fore owners’ individual value judgements and ethics. For example, owners may not feel that they need a formalised tool or checklist to help them monitor aspects which they already think about (a finding synonymous with the experiences of professional riders [6]). However, owners’ existing welfare assessments are very much based within their own frame of reference; therefore, sharing resources, case studies or other owners’ experiences which broaden that range of reference or challenge existing heuristics, could help owners to consider new ideas.

Owners felt a responsibility to make appropriate decisions and when doing so they made, or questioned, judgements about what was considered to be “right” or “wrong”. In this study, people’s approaches to horse care changed over time and so did the circumstances in which they considered their horse’s condition or management to necessitate a concern. Our findings highlight the network of influences which shape a person’s ethical views; one major influence was the social groups which shaped their beliefs. Taken-for-granted norms about horse-keeping practices and what constitutes ‘optimal’ equine wellbeing are established and maintained through equine communities and these can negatively impact horse wellbeing. Elsewhere Hausberger et al. [33] discuss how traditional practices such as bit use “are rarely questioned” and so practices which negatively impact equine welfare are repeatedly used. Our findings show that where owners do question certain practices this can prompt them to seek information or social support from elsewhere—for instance through online groups.

In canine health, researchers have argued that those seeking to improve dog welfare need to look beyond increasing owner knowledge through education, by considering the complexity of real life interactions with owners and dogs and incorporating internally held attitudes, values and beliefs around welfare [34]. Equine research has found that owner knowledge of issues that could compromise welfare did not necessarily translate into practical application of this knowledge when owners made decisions for their own horse [35]. Our findings demonstrate how owners assessed a horse’s wellbeing through a lens generated by a number of individualised factors. It was through this understanding that management decisions were made. Principles of “good” horse care were negotiated based upon individual relationships with a horse and knowledge of them over time, demonstrating where supposed ‘discrepancies’ between this knowledge and practice can arise.

In our study, owners also used heuristics as a strategy to assess what was right or wrong about a situation, they found it easier to apply those heuristics to others’ horses as compared with their own situations. As well as assisting owners in increasing knowledge about affective states, we consider that encouraging owners to explore their taken-for-granted value judgements could help them to make informed decision about their horse’s wellbeing and needs. Providing support which accommodates for multiple perspectives on horse-keeping practices and acknowledges these conflicts that arise will be important to their applicability to practice.

Veterinary advice in relation to equine QoL was infrequently discussed in this study. Nevertheless, assessing a horse’s QoL is considered a core part of clinical decision-making for veterinarians [36], and owners of older horses have reported both quality of life and veterinary advice to be important in end-of-life decision-making [37]. Therefore, better communication around this important topic is needed. Where veterinarians are consulted, the findings highlighted in this study will enable a better understanding of how owners construct and communicate their assessment of the horse. By focusing on the context in which language associated with QoL is used, veterinarians and others professionals involved in care can better communicate and support decision-making in practice. Further research into the language used by veterinarians when discussing equine quality of life is also required.

In this study, owners felt that euthanasia should result if they could not maintain the horse within their construct of an acceptable range of wellness and it was generally considered a responsible course of action. However, descriptions of issues requiring euthanasia varied dramatically between participants and findings highlight the very individual nature of deciding what is, or is not, an acceptable life for an animal. Owners’ principles were contextualised as specific situations arose and therefore highlight how decision-making is shaped by individual horse-human relationships. In order to explore how different horse-human relationships shape approaches to end-of-life decision-making, further research into the perspectives of other equine caregivers and professionals is needed.

This study used qualitative research methods, specifically focus groups, to explore horse owners’ experience; while there are numerous benefits to this choice, there are also limitations. Firstly, participants who took part in the focus groups may have represented those with a particular interest in, and time to consider, equine welfare. Future studies could aim to include a convenience sample of more diverse perspectives, as well as comparing horse-keeper perspectives with those of other stakeholders such as veterinarians and livery yard managers. Secondly, this study explored owners’ perspectives, but those may not always match the behaviours people enact in real-life settings, and additional ethnographic or observational methods would be required to compare the dilemmas around welfare that owners describe with their actions and experiences in real-time. Finally, it has been documented that focus groups lead participants to form consensus opinions [38,39]. While we were aware of this limitation, and aimed to encourage diverse opinion and dissent, future individual interviews could supplement our dataset in order to allow participants to share their experiences without the potential for group consensus forming.

## 5. Conclusions

This study has demonstrated that horse owners did not clearly delineate between the terminology associated with equine wellbeing. Rather, we found during focus group discussions terms were used by participants to express their experience of making ongoing, informal assessments of their horse. Knowledge produced through individual horse-human relationships generated capital, and this contributed to owners’ understanding of wellbeing both in terms of what was necessary to provide for a horse and in how to interpret the horse’s experience.

These findings also demonstrate how terms were employed by participants when expressing judgements about whether horse-keeping practices were ‘right’ or ‘wrong’. Constructs of wellness were fluid, and in certain situations, necessitated compromise when managing their own horse. While participants used wellbeing related terminology interchangeably, the context of discussions about horse-keeping led to a mutual intelligibility. It was not the terms themselves that were important in describing wellbeing between participants, but their context.

The use of relatively reductionist approaches such as tick lists or welfare checklists may seem obsolete to owners, who already feel they are assessing wellbeing on a daily basis. Rather than focussing on basic indicators, such as appetite or lameness (which were already considered in informal assessments), instead, owners relied upon other factors they considered to be important, such as their knowledge of the horse or ‘capital’. We suggest that encouraging owners to explore their own attitudes, values, and beliefs will have merit in changing perspectives around issues that impact equine welfare, particularly coupled with resources which broaden their experience of welfare assessment decision-making.

## Figures and Tables

**Figure 1 animals-12-02937-f001:**
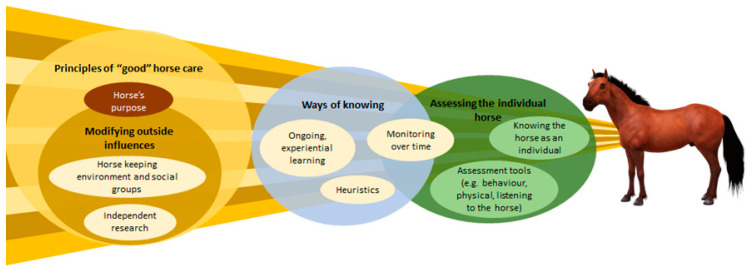
Conceptual model representing themes identified which shaped leisure horse owner’s understanding of a horse’s wellbeing.

**Table 1 animals-12-02937-t001:** Focus group participant details, including a summary of horse owning experience.

Focus Group	Sex	Horses
1	F	3 horses kept at privately rented field, one is ridden
	F	2 horses and companion pony kept at home, one is ridden
	F	Riding instructor and coach, does not currently own a horse of her own
2	F	3 horses kept at livery; 2 competing in riding club competitions
	F	2 horses kept at home; one semi-retired, one fully retired due to injury
	F	2 horses kept at livery; one retired and one competing in dressage, shared with her daughter; also shares another person’s horse 2 days a week
	F	The daughter of participant above; competes in riding club competitions
	F	Two horses kept at livery, one retired due to injury, the other competing in eventing
	F	One horse, kept at livery, currently being broken in
3	F	One horse kept in a friends’ field, used primarily for hacking
	F	One older horse kept at livery, used for hacking
	F	One horse kept at home, previously evented but now semi-retired
	F	Four horses kept at home, including an in-foal mare. Two previous event horses now retired, the fourth mainly hacks
	F	Two horses kept at livery, competing in showjumping
4	M	Four horses kept at home; used for carriage driving
	M	Shares his partner’s horse, kept at livery
	M	Shares his partner’s horse, kept at livery: has previously shared racehorses in training as part of a syndicate
	M	Four horses kept at home; used for hacking
	M	Three horses kept at home; one companion, the other two compete (one belongs to his wife), take part in fun rides
	M	Two horses (one belongs to his wife), both take part in endurance competitions and fun rides, kept at home
	M	One endurance competition horse, four companion ponies, and a mule; kept in private field

**Table 2 animals-12-02937-t002:** Results of a content analysis of the terms of interest.

Item	Number of Instances	Meaning	Number of Instances	Examples
Welfare	56	Ambiguous—General sense of wellness	32	“*we’ve been monitoring our guys welfare. My wife’s horse is coming back from a tendon injury that he did last summer, so we’ve been doing some rehab with him.*”* note: many participants used the terms “welfare” and “wellbeing” concurrently as in “*you’re constantly thinking about the welfare and wellbeing of your horses*”
		An outcome (positive welfare)	3	“*But it is so worth it to see that welfare come out of him. It is that good welfare that he has got there in the care by somebody else.*”
		Basic acceptable standards provided	9	“*every day I pass these two ponies in a field that isn’t a great field. They have a field shelter, they don’t get ridden. I see people there every day, so they get water, but they don’t look well cared for…they have fresh haylage, fresh water. But for me, I don’t think it’s a good life for the horses. But I do think that their welfare has been met.*”
		Minimum acceptable quality of life	6	“*it was obviously in pain. It was obviously uncomfortable. That, to me, isn’t welfare.*”
		Provision	5	“*When we talk about welfare, for me that equates to human intervention, and if I am doing something to affect that horse’s welfare in a way that restricts its natural way of living, that’s welfare.*”
Wellbeing	22	Ambiguous—Provisions to ensure a general sense of contentment or health	19	“*I even come to the point of thinking,* “*Has he got the right rug on at the right time, is he going to be cold?*” *Which might sound irrational, but I think it definitely does, and that comes into their wellbeing.*”
		Basic acceptable standards	2	“*those fundamentals that shouldn’t change through a horse’s life so that the basics for wellbeing, I read actually your summary and I’ve never heard it described as the three-Fs. But it’s that, isn’t it? It’s from when a horse is born that it should have those fundamentals through its life*”
		Minimum acceptable quality of life experienced by the horse	1	“*welfare and wellbeing, it obviously worked out she had another three years fine, but if it ever happened again, I don’t think we’d do that again, would we?*” [in relation to an older horse needing to be hand-walked due to an episode of painful lymphangitis]
Quality of life	30	Positive experiences	7	“*If we go off to the beach, if we go off to a fun ride, she really enjoys it. She keeps up with all the others, even goes past some of the others. And I think that is a nice quality of life.*”
		Minimum standards; is the life worth living?	19	“*we identified a severe underlying problem. And it was down to quality of life. This is a horse, it has got to be able to graze in a field, run and play, and skip and joy or whatever. And that horse would never do that.*”
		Ambiguous—a sense of wellbeing	4	“*you start to put a measure against those points, and think,* “*Well, is he happy? Is he having a good quality of life?*” *His mental health may be there.*”
Best life	23	An aspiration	8	“*I want them to have the best life that they can that I’m able to give them.*”
		Optimum conditions, or experience, for the horse	12	“*a horse can have a good life but a horse’s best life, it certainly wouldn’t ever been ridden or has to do anything for humans….*”
Good life	10	An adequate, satisfactory experience or provision	10	“*I don’t think it’s a good life for the horses. But I do think that their welfare has been met. But is it the best life? No*”
Life worth Living	14	Minimum acceptable experience or provision	7	“*When you think about a life worth living, I think the horses that have the worst life are things like racehorses, they don’t have a life worth living. They really don’t. They don’t have anything a horse needs, the top show jumpers and especially the top dressage horses, they don’t have the life worth living at all.*”
		Positive experience	2	“*do I think my horse has a good mental health? And if he does, and he probably has a good life. He’s probably happy, it’s a life that’s worth living.*
		Questioning moral judgements about horse experience	5	“*A life worth living. I don’t think horses are that existential. (Laughter) When somebody says,* “*Oh, this horse is wasted on me,*” *it is like they don’t cry themselves to sleep because they are not winning championships.*
Happiness	65	Positive affect	6	“*I saw him being in a field, a massive field, with three other horses that my friend has, and he is just so happy. You can just tell. He is a different horse.*”
		Contentment	6	“*I think if you want to know if your horse is happy, look for contentment. You know, are they chilled out when they’re just roaming around having a munch?*”
		Absence of happiness	14	“*People used to say to me she’s really not well, she’s not happy, you should have her put down.*”
		Horse’s general acceptance of conditions; general wellbeing	39	“*he’s quite happy to be turned out most of the time, and I am lucky enough to have access to plenty of grass and some stables at home.*”

## Data Availability

Anonymised data can be shared upon request with research organisations, subject to ethical approval. Please contact the authors for more information.

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
