# Peer review of "A Qualitative Exploration of UK Leisure Horse Owners’ Perceptions of Equine Wellbeing"

_animals, 2022, doi:10.3390/ani12212937_

Round 1
Reviewer 1 Report
Is the title meant to be 'Quine' or 'Equine'?
Mention grounded theory in abstract, and the four major themes developed.
Any 'official' definitions of the different terms assessed?
Line 244, Line 714 'didn't change to 'did not'
Line 273 'ideal conditions' - but the research indicates this does not exist. Need to discuss and evidence what is 'ideal' if looking to highlight differences from this ideal.
Line 654 - 655 double negative
Line 743 isn't to is not
Interesting that the participants note the individual knowledge in assessing wellbeing, yet make judgements about other equines that they see regularly - but don't know in the way they know their own equines.
A really interesting study and well-written paper. If an overview of some of the key 'takeaways' from Table 1 can be included that would be great, as that lends weight to the discussion around the usage of the words for 'wellbeing'.
Reviewer 2 Report
Dear authors,
Please see the recommendations, corrections, and questions formulated during the revision process of your manuscript and use them as considered to increase the value of your paper.
The idea and subject of the manuscript are innovative, timely, and very important. However, the paper needs a thorough edit, to enhance clarity and conciseness; also, its inherent structure has to be improved – to follow a logical sequence in each section. Some supplementary files should be moved in the manuscript, as this has to be comprehensive and fully understandable without the supplements (e.g., tables in the Results section).
Title
Please correct ‘quine’ to ‘equine’
The title is correct, yet it may come across as awkward to the reader – it is about the exploration of the owners’ perception regarding their horses’ wellbeing, not the actual horse’s wellbeing in the owner (that is impossible), please consider a possible improvement in this regard.
Simple summary
L13-14: Please clarify to whom this sentence refers. If it regards the horse owners, please specify, as the terms are not ill-defined in the specialty literature (even if interchangeable use happens at times).
L17-18: please consider changing the word order to ‘Individually constructed equine wellbeing assessments were ongoing and shaped…’
L19: What do you mean by ‘heuristics’ of the horse, and how that links to its usage? For a simple summary please consider using simple and clear phrasing. The same applies to L20.
Abstract
L22: please consider using ‘still not fully defined’ or similar, instead of ‘ill-defined’ – the latter suggest intentionality (in my humble opinion)
L25-L38: A suggestion for more fluency, repetition avoidance, and word count reduction would be: ‘This study sought to establish how UK leisure horse owners use wellbeing-related terminology by exploring their accounts within a focus group setting. The four online focus group discussions (FGD) selected for qualitative data collection involved a semi-structured discussion, followed by a group activity to compare seven equine wellbeing-related terms of interest introduced by the facilitator. The collected (or extracted) data were analysed using a constructivist grounded theory approach, and also by content analysis, to examine the frequency and way of usage (or ‘subjective meaning’) of the terms of interest. The results showed (or ‘The analysis identified’) that horse owners did not clearly delineate between different terms, rather, they used variably (or ‘interchangeably’?) the terms in the context of their own assessments of their horses. Their way to assign meaning to what they saw in their horses was individual and subjective, shaped by past experiences, relationships with their animal, and peers or social groups. This individualised construction of equine wellbeing terms impacted how the owners described their approach to horse care. In this study, we aimed to extend the literature on equine wellbeing terminology usage and to highlight differences between the academic literature and the real-world experiences of horse owners.’
Introduction
Please revise and correct the in-text citations to fit the Journal’s guidelines (numbers in square brackets instead of superscripts)
L60-62: please move the citation [9] after ‘A recent study’. L61: please change ‘in the abstract’ to ‘in theory’ or ‘on a theoretical level’ (avoid misleading reference to the specific study’s abstract / summary); change second ‘people’ to ‘they’
L64: consider to change ‘at times when views around a horse’s welfare or QoL were’ to ‘when horse welfare or QoL views were’, delete ‘exactly’
L67: consider changing ‘language’ to ‘terminology’; this first sentence is rather redundant
L68: consider changing ‘expressions of wellness’ to ‘wellness-related terminology’ and ‘terms’ in L69 to ‘denominations’
L71-72: please rephrase to avoid repetition ‘the importance… as important’
L76: please move the citation after ‘Recent research’
L94-95: remove ; after ‘sought to’; change ‘people’ to ‘horse owners’; consider changing ‘language’ to ‘terminology’ (equine wellbeing terminology’ instead of ‘language in relation to equine wellbeing’); insert ‘the’ after ‘understand’; consider to change ‘collective sense-making’ to ‘collective understanding’
Materials and methods
Please reorder and reorganize the information presented in this section to follow a logical continuum and to make the study as much as possible repeatable (e.g. include clear selection criteria of the participants, give the area of the country instead ‘a range of locations’, state the range of participant ages, and so on). The first sentence does not fit the section, it is either redundant or can be considered a result statement. The logical progression of the section would be, in my opinion:
1. qualitative data collection period,
2. participant selection criteria, participant agreement insurance,
3. description of the total study sample (total number, age-range, gender, location area, a slightly better description of their horse-owner experience—for how long did they own horses generally, number of horses owned, housing system used for their horses and if they kept their horses at their domicile or not—any relevant information of this kind, if available),
4. criteria for including the participants in FGs, description of the FG—the fourth one included, here, at the section beginning, no of participants/FG
5. description of the FGDs (planning included): methodology&technical description (zoom, 90 mins, recording, facilitators), anonymization of participants, reason, and description of the 2 parts, the horse-owner quality of the facilitators and the data drawn from it,
6. post FGD processing of data
7. data analysis to obtain results.
Results
L143-147: these are rather conclusions than results. However, consider moving these at the end of this section, if not in the conclusions section. The sentence in L156-157 is redundant, please do not state what you are going to present, just present it. Consider changing it to ‘Besides the exploration of linguistic constructs of horse owners about the wellbeing of their animals, our data analysis provided results also about how the participants’ understandings on specific terms have been generated’ or similar.
L165-166: please consider changing the usage of quotation marks (to “principles of ‘good’ horse care”, “modifying outside influences” and so on, here and hereafter)
L166-167: please insert (Figure 1) after ‘assessing the individual horse’ and delete the next sentence (‘These are presented…’)
Figure 1 caption: please complete it, such as to be understandable as a stand-alone for the reader (insert ‘of the studied leisure horse owners’ or similar)
L182-184: please change the phrasing. At the baseline, the reader would consider that a veterinarian (please consider changing ‘vet’ to ‘veterinarian’ or ‘veterinary professional/practitioner’ throughout the manuscript) is able to perform a more objective and better-informed welfare assessment of the animals. Thus, the owner should be compared to the veterinarian and not vice-versa, if you are going to keep the comparison with a veterinary professional—or remove the exemplification from within the brackets/change it to ‘another horse owner’. The veterinarian’s perspective, besides ideology and experience, shall be based on objective knowledge too, which is not an aspect to be overlooked. Practically I would suggest either ‘Consequently (avoid ‘therefore’ repetition) these horse owners might have an entirely different perspective on the wellbeing of their individual horse, based on individual experiences and ideologies, compared to the assessment performed by another person, such as a veterinarian’ or keep the sentence (eliminate ‘therefore’ repetition) and replace ‘a vet’ with ‘another horse owner’
L175-189: please reduce wordcount, eliminate redundancy and repetition
L190-193: please rephrase, do not announce in advance what you are going to present.
L195-198: change ‘For example' to ‘e.g.’ and do not separate these 2 sentences
L198-208: how can we know that wider equine communities, such as groups (assumably social media, which should be stated if that is the case) had been the influencers of these owners? Naturalness is widely discussed in animal welfare scientific literature; we could claim that as an influence source for them. Also, the phrasing suggests totally informal groups, in a slightly disapproving way, maybe because of the use of quote marks. In this regard we should not forget about reliable research entities, professional charities, and so on (especially in the UK), who disseminate relevant and research-based information/educational materials through social media; the need for naturalness in horse management could be very well in their topics. On the other hand, a major indication of blending horse needs and own identity of the owner (described in L2014-208), not just as a horse owner but as a person, appears right in the quote given, should you wish to highlight it (and make a more organic transition to the next lines)? Describes management (out, no rugs, no clipping, bitless, etc) AND reasons about oneself “natural is just MY THING. I TRY TO LIVE AS MUCH AS A NATURAL LIFESTYLE I CAN” — the own lifestyle and orientation is determinant, more than the horse’s perceived needs
L241: consider changing ‘In the abstract (for example…’ to ‘When discussions were purely theoretical (such as discussing…’ (avoid repetition)
L242: incomplete sentence (merge them)
L243: correct ‘Participant’s’
L244: please do not use contractions in academic writing (quoting study subjects is different)
L245: insert ‘about’ after ‘discussed’
L243-250: it is not clear whose quote is this of the 2 participants mentioned, and ‘described drawing on these principles’ should be reworded for more clarity too
L251-254: please consider adapting to academic writing the use of singular and plural
L255: replace ‘this’ with ‘it’ and ‘in reality, perceptions of equine wellbeing could differ’ to ‘when inspected/considered/ evaluated objectively, the perceptions of equine wellbeing were different from a participant to another’
L256: consider to change ‘Similar inter-participant differences have been revealed about the moment at which…’
L259: delete ‘to become’. What do you mean by ‘over time’? please specify
L261: incorrect punctuation (;)
L263-265: change ‘around’ to ‘on’, correct ‘of an individual horses’ wellbeing’
L269-271: consider to change ‘around’ to ‘about’, delete ‘in reality, change ‘in’ to ‘regarding their horse’s…’ In my opinion, the compromises regarding management changes of the horse doesn’t necessarily imply changes in the owner’s perception about their horse’s wellbeing. If their perception changed, they would not consider a compromise as such (but they would consider it the right decision for a better wellbeing). Remove comma after ‘yards’
L273: keep time consistency (might not have been possible)
L274-275: please use en dash or em dash instead of the hyphen
L329-330: please change wording to avoid repetition (‘result, result, resulted’)
L340-341: just to double check, one horse owned by several partners? For added clarity, consider inserting ‘better’ after ‘therefore’
L409: mention in brackets not needed
When general descriptions are given, such as within the 3.1.4. subsection (but other parts too),l467 it would be beneficial to state the number of percentage of participants whose beliefs were similar about certain topics (and how many of them did disagree)
L509-513: consider rephrasing by removing discussion elements from the result section
L517: ‘could be’ or were?
L517: reword properly ‘displayed in see supplementary item’
In my opinion, restricted tables with results should be given in the main manuscript.
L530: add ‘The term’ before ‘Welfare’
Discussion
L623: please change ‘muddiness’ to a clearer term
L625: please keep tense consistency (past)
L634-636: very good point!
L663-666: please move the citation after ‘previous studies’ and give more than one (or change ‘studies’ to singular)
L693: citation needed (please repeat it here)
L705: please move the citation’s number after the mentioned citation
L742: please keep tense consistency (past), also in L744
Conclusions
L766-767: please avoid repetition (generated)
L770: no need for mentioning canine studies (or citation is needed)
As stated in the Discussions section, despite unclear delimitations of the terms used, the FG participants found mutual understanding for the terminology used—please insert a mention about that in this section
References
Please format this section according to the Journal’s guide.
Author Response
Many thanks for your comments, we particularly appreciated that you had taken the time to suggest alternatives rather than simply criticising what was there – that is very kind and was appreciated. Thank you for your valuable feedback, and we hope we have now substantially improved the paper as a result. Please see the attachment for detailed responses.

Reviewer 3 Report
The manuscript provides a useful insight into the perception of ‘wellness’ and related terms by leisure horse owners, with an interesting discussion on how an understanding of this by the owners themselves could prompt improved understanding of welfare, and by the wider equine community could shape the support that equine owners receive. I enjoyed reading the manuscript and feel that the authors have done an excellent job of untangling a particularly nuanced topic.
I have made a number of suggestions for minor changes below:
Abstract and intro
Line 50: Suggest a change to “…research around human perception of equine wellbeing has focused on equestrian professionals…” or similar
Line 53: Was the research in the previous line UK specific as well? It would be good to specify this to make the comparison sound.
Line 56: Specifically leisure horse owners? (perhaps a ‘thereafter referred to as ‘horse owners’ after the first instance if so)
Line 64: Was it that views were divergent (i.e. difference of opinion about what constitutes ‘good’ welfare), or that terminology was divergent (i.e. end goal may have been the same, but not communicated effectively)? The sentence citing reference 10 seems to imply the former, but the following text implies the latter. Could this be clarified?
Line 70: I can see how this is difficult to tackle. I wonder if a definition of ‘welfare’ or ‘wellbeing’ as you have decided to use them in the introduction would help? There have been quite a few terms used so far (health, welfare, wellbeing, wellness, QoL) – I’d be interested in how these terms have been used or defined in the literature thus far, which would provide context for the current study. It might also be helpful to limit the number of terms used in the intro / descriptive writing when not specifically referencing a result (could you use ‘wellbeing’ wherever possible to avoid ambiguity?)
Line 74: Would the Qualitative Behavioural Assessment (QBA) be considered a QoL assessment? See Wemelsfelder F. (2001) doi: 10.1006/anbe.2001.1741
Line 83: I found this paragraph a little clumsy. Line 83 seems to contradict line 89. I suspect a slight reword can clarify.
Line 141: Were any further ethical procedures followed (i.e. the right to withdraw etc.)?
Results
Line 206: Move “(without shoes)” to the first instance of the phrase “barefoot” on line 197
Line 221: It isn’t clear from the quote whether the participant was suggesting these restrictions are acceptable / a concern, or if they are simply observing them – is there another quote to demonstrate the text from line 218?
Line 389: Were participants asked how they developed these beliefs (e.g. box rest is not good..)? They are described as self-created but this accurate?
Discussion
Line 642: Suggest changing to ‘social capital’ (including references to capital throughout) if this is an accurate term
Line 658: The results state that participants didn’t use the word ‘happy’ very often, but I’m not sure this is enough to conclude that they disliked the word. Use of the term ‘not happy’, specifically, isn’t mentioned in the results – can this be added please?
Line 692: Suggest giving the reference here instead of ‘aforementioned competition horse study’
Line 703: Following my earlier comment about definitions, I feel there needs to be some explanation (here or in the intro) about what is generally accepted within the animal welfare community as ‘good’ or ‘optimal’ wellbeing or welfare. Otherwise the discussion about how owners perceptions of wellness impact ‘actual’ wellness (which seems to be the point of this statement) start to sound circular.
Line 718: Do you mean discrepancies between knowledge of what constitutes good welfare and the practical application of this knowledge? How would highlighting how principles of ‘good’ welfare are negotiated show where discrepancies occur?
Conclusion:
Line 777: Suggest changing the wording to “instead, owners relied upon other factors, such as their knowledge of the horse…” or similar
I feel this section could be streamline slightly. The authors could consider the removal of phrases such as ‘This study has demonstrated…’ etc. for brevity and ease of reading and dive straight into the text, depending on their preferred writing style, of course.
Author Response
Thank you for your valuable feedback, and we hope we have now improved the paper as a result. Please see the attachment for detailed responses.

Round 2
Reviewer 2 Report
Dear authors, I do consider that your paper improved considerably compared to its first version submitted. Thank you for your replies, for the opinions expressed and corrections made. My comments regard only a few minor details, please find them below:
R1: Please revise and correct the in-text citations to fit the Journal’s guidelines (numbers in square brackets instead of superscripts)
Authors: amended
R2: No, you did not
End of introduction, L111-112: ‘Early detection of health issues may prevent more serious issues from developing, but in order to detect these issues, it is important that horse owners have an understanding of their horse’s baseline and positive welfare indicators. Furthermore, as scientific research has focussed on specific health issues, there has been a paucity of research around how horse owners conceptualise, monitor, and manage good health and wellbeing in their horses – or the language 116 they use to describe this.’ Please correct the repetition of ‘issues’.
Figure 1 caption: Please move the caption below the figure (I apologize for not requesting this in my first review)
R1 on L198-208
R2: Nothing to be corrected, comment on the Authors’ reply: We all know that your point of view is true about the influence of social groups, my first review comment was because of scientific soundness: we cannot state scientifically unproven concepts (and to prove those is outside the aim of this research). As for the rest, I (personally) would have liked the deeper explanation of this owner's psychology/views... but I do agree with you--again, not in the scope of this paper.
R1 on 3.1.4. subsection
R2: Nothing to be corrected, comment on the Author’s reply: You are correct with this, I do understand and agree. Excuse the more quantitative-approach used mind.
Reference list: Dear authors, we all hate to format our reference lists.
1. If you used the Animals template, you had there the exact examples for formatting each type of reference.
2. I do not find your list consistent. Some titles have all authors given, others only the first, followed by 'et al.' (which I have never seen in any serious reference list); some titles have doi, others do not; some have page numbers, others do not; and so on. When the Journal refers to a style, I assume they mean one of the consecrated citation styles (APA, MLA, Chicago, etc); it may exist one with the publication year given in brackets at the very end of the citation, but I did not come across that to date. It is not a major issue in any regard, but I do think that we all have to comply to the Animals Journal format in this regard.
Author Response
Thank you for your additional feedback. We have made some final amendments to the paper - please see the attachment for our responses.
